# β1-Integrin-Mediated Uptake of Chondrocyte Extracellular Vesicles Regulates Chondrocyte Homeostasis

**DOI:** 10.3390/ijms25094756

**Published:** 2024-04-26

**Authors:** Mohammed Tayab Hussain, Shani Austin-Williams, Thomas Dudley Wright, Umesh Kumar Dhawan, Andreia L. Pinto, Dianne Cooper, Lucy V. Norling

**Affiliations:** 1The William Harvey Research Institute, Barts and the London School of Medicine and Dentistry, Queen Mary University of London, Charterhouse Square, London EC1M 6BQ, UK; m.t.hussain@qmul.ac.uk (M.T.H.); t.d.wright@qmul.ac.uk (T.D.W.); u.dhawan@qmul.ac.uk (U.K.D.); d.cooper@qmul.ac.uk (D.C.); 2Royal Brompton & Harefield NHS Foundation Trust, London SW3 6PY, UK; a.pinto@rbht.nhs.uk; 3Centre for Inflammation and Therapeutic Innovation, Queen Mary University of London, London EC1M 6BQ, UK

**Keywords:** osteoarthritis, chondrocytes, extracellular vesicles, β1-integrin

## Abstract

Osteoarthritis (OA) is the most prevalent age-related degenerative disorder, which severely reduces the quality of life of those affected. Whilst management strategies exist, no cures are currently available. Virtually all joint resident cells generate extracellular vesicles (EVs), and alterations in chondrocyte EVs during OA have previously been reported. Herein, we investigated factors influencing chondrocyte EV release and the functional role that these EVs exhibit. Both 2D and 3D models of culturing C28I/2 chondrocytes were used for generating chondrocyte EVs. We assessed the effect of these EVs on chondrogenic gene expression as well as their uptake by chondrocytes. Collectively, the data demonstrated that chondrocyte EVs are sequestered within the cartilage ECM and that a bi-directional relationship exists between chondrocyte EV release and changes in chondrogenic differentiation. Finally, we demonstrated that the uptake of chondrocyte EVs is at least partially dependent on β1-integrin. These results indicate that chondrocyte EVs have an autocrine homeostatic role that maintains chondrocyte phenotype. How this role is perturbed under OA conditions remains the subject of future work.

## 1. Introduction

Osteoarthritis (OA) is a chronic degenerative disorder of the joint associated with age, which can cause debilitating alterations in joint mobility, long-term pain, and an overall decrease in patients’ quality of life. Having OA is associated with an up to 1.6-fold higher risk of all-cause mortality than the general population [1]. OA is typified by hallmark anatomical changes to affected joints, including cartilage degradation, sub-chondral bone thickening, osteophyte formation and, eventually, hypertrophy of the joint capsule. Changes in gross anatomy are underpinned by molecular mechanisms [2]. Briefly, cartilage destruction can occur following traumatic injury or chronic wear and tear. The destruction of cartilage can cause the death of chondrocytes within the lesion and the release of extracellular matrix components as damage-associated molecular patterns. These events induce an increase in pro-inflammatory signalling and cytokine release in the surrounding cells. In part, this can lead to an increase in the expression and activation of matrix-metalloproteinases (MMP), such as MMP-13, which further exacerbate cartilage breakdown. Alongside this, increases in pro-inflammatory signalling also reduces the production of cartilage components from chondrocytes which have not undergone death [3]. Overall, a decrease in negatively charged cartilage proteoglycans results in decreased retention of water within the tissue [4], and subsequently increases tissue stiffness. The outcome of this is a reduced ability of the joint to withstand compressive loads [5]. Greater cartilage stiffness reduces chondrogenic gene expression and puts the individual at risk of future injury [6]. Collectively, these processes, repeated over a sufficiently long period of time, compromise the balance of catabolic and anabolic mechanisms regulating joint homeostasis and eventually present as OA. Alongside age, post-traumatic joint injury and hormonal changes, typically associated with menopause, dramatically increase the risk of an individual developing OA [7]. Unfortunately, to date, there are no long-term curative therapies available in the treatment of OA. However, an area of growing pre-clinical work in this regard is the use of extracellular vesicles (EVs) as a tissue regenerative therapy [8].

EVs are biological nanoparticles consisting of cargo either encapsulated within or presented on the surface of a lipid membrane shell. EVs are mediators of paracellular communication and are released constitutively from virtually all cells of the body [9]. EVs’ size dictates their classification, with the smallest EVs being classed as exosomes, intermediate-sized EVs (150 nm–1 µm) as microvesicles and the largest as apoptotic bodies. Each classification of EV is underpinned by a unique mechanism of biogenesis. For example, microvesicles are generated via the outward blebbing of the plasma membrane of cells in response to changes in intracellular calcium levels. In contrast, exosomes are generated following the inward budding of the endosomal membrane in the luminal space of multivesicular endosomes undergoing maturation [10]. Importantly, EVs are heterogeneous, and their contents include a range of proteins, cytokines, nucleic acids and lipid mediators. The proportion of EVs carrying a certain cargo, whether the cargo is encapsulated or presented on the EV surface, and the amounts of exosomes versus microvesicles is wholly dependent on the phenotype of the cell of origin. Generally, EVs generated from a pro-inflammatory cell will have pro-inflammatory effects and vice versa [11]. However, there are exceptions to this; for example, EVs from TNF-α stimulated neutrophils can exert a pro-resolving effect via Annexin-A1 in the context of inflammatory arthritis [12]. There is good evidence for the functional effects of EVs in a range of homeostatic processes as well as in a variety of diseases and pathological conditions.

EVs from a variety of sources have been used in pre-clinical trials to attenuate the osteoarthritic phenotype. These tools offer advantages over conventional therapies, for example, the ability to promote targeted delivery and the protection of therapeutic payloads from enzyme-rich microenvironments (Reviewed in [8]). The limitations of cell-based therapies include poor localization to the site of interest and an inability to control cell fate in the diseased microenvironment. MSC-EVs are able to promote a greater resolving phenotype than the parent cells in a model of murine OA [13]. In fact, in a metabolically induced model of rat OA, MSC infusion further exacerbated cartilage degradation whereas MSC-EVs did not [14].

Exogenously administered chondrocyte-derived EVs have a potent pro-chondrogenic and protective phenotype in OA. These EVs have been shown to promote type II collagen and aggrecan production in chondrocyte cultures and attenuate the extent of synovitis in a mouse model of OA [15,16]. Endogenous chondrocyte EVs are present within the cartilage and the proteome of these EVs becomes altered during OA [17]. Both Liu et al. and Ni et al. have shown that EVs from IL-1β-stimulated chondrocytes promote catabolic gene expression in recipient healthy chondrocytes and exacerbate cartilage degeneration [15,16]. Current data suggest that chondrocyte EVs are detected in articular cartilage and these microstructures may regulate chondrocyte phenotype in health and disease [11]. However, their exact role, mechanism of action and how this becomes perturbed in the disease state are poorly understood. Herein, we aim to interrogate the factors that may influence the biology of EVs from chondrocytes, how this may change during the pathogenic shift from health to disease and why this is relevant to a chronic degenerative disorder like OA.

## 2. Results

### 2.1. Characterisation of Chondrocyte-Derived EVs

We first compared chondrocyte EV generation utilizing two different methods of culture, monoculture (2D) or pellet culture (3D), as it is known that 3D culture formats generally promote greater chondrogenic gene expression when compared to cells in 2D culture [18]. Chondrocyte growth in 3D culture also permits extracellular matrix deposition and therefore more closely resembles a normal chondrocyte microenvironment [19]. Surprisingly, the 2D cell culture yielded significantly higher numbers of EVs in comparison to 3D culture supernatants, as assessed by nano-flow cytometry (Figure 1A). We hypothesized that the low yield of EVs from 3D cultures could reflect the trapping of EVs within the more extensive extracellular matrix. To investigate this, EVs were isolated from cell culture media either before or after the mechanical dissociation of 3D cultures. As before, the number of EVs isolated from the cell culture media in the absence of dissociation was relatively low. In comparison, the mechanical dissociation of 3D cultures resulted in an increase in the number of EVs that were quantified (Figure 1B). Therefore, we directly compared the number of EVs isolated from either the supernatant of chondrocytes in 2D or 3D following dissociation as two model systems for generating chondrocyte EVs in vitro. The subsequent yield of EVs from 3D cultures proved to be significantly greater than those isolated from 2D culture supernatants (Figure 1C).

We further characterized these EVs by transmission electron microscopy (TEM) according to the standards outlined by the Minimal Information for Studies of Extracellular Vesicles guidelines (MISEV) to provide further validation that the mechanical disruption of 3D cultures did not damage the EVs or lead to cellular debris. Samples from 2D and 3D cultures were imaged using electron microscopy, with and without detergent treatment, used to disrupt EV lipid membranes (Figure 1D–G). The vast majority of EVs were observed to have a single lipid bilayer membrane with a round shape. The treatment of either type of EVs with 0.01% triton-X (Figure 1E,G) led to a significant reduction in viable EVs relative to untreated samples (Figure 1D,F). A random selection of 15 fields of view per sample were imaged and EVs were sized. Based on electron microscopy analysis, there was no difference in the size profiles of EVs isolated from 2D or 3D cultures, as seen with the nanoFCM (Figure 1H,I).

To further define the extracellular vesicles generated by 3D chondrocyte cultures following dissociation, and in line with the MISEV guidelines, cell pellets and EVs were analysed via Western blotting [20]. Lysates from chondrocytes and EVs were probed for the tetraspanin CD63, and EV purity was assessed by the ‘exclusion marker’ calnexin, which is cell-specific and expected to be absent in EVs. As anticipated, Western blotting confirmed the presence of both CD63 and calnexin in cell lysates and only CD63 in EV lysates (Figure 1J,K). Collectively, these methods validate the protocol applied to isolate EVs from 3D pellet cultures, as well as the genuine vesicle nature of these structures.

### 2.2. Effect of Cytokine Stimulation on Chondrocyte EV Release

Having optimised the workflow for chondrocyte EV generation, isolation and quantification, the effect of cytokine stimulation on EV release from 2D and 3D cell cultures was next investigated. We tested the effects of IL-1β, a catabolic stimulus for chondrocytes, and TGF-β, a promoter of chondrogenesis. The concentration of EVs isolated from 2D chondrocytes was unaffected by either cytokine treatment (IL-1β or TGF-β) with respect to untreated controls. Similarly, the treatment of 3D cell cultures with IL-1β had no impact on the EV yield; however, treatment with TGF-β led to a significant increase in EVs collected from 3D cultures (Figure 2A). We also observed greater ECM accumulation in 3D cultures stimulated with TGF-β compared with vehicle control using Alcian blue staining (Appendix A). The yield of EVs from 3D cultures was greater than that of 2D cultures (Figure 1C) and remained higher in all treatment conditions (Figure 2A).

To validate the chondrocyte response to the cytokine treatment, the expression of anabolic (*SOX9*, *COL2A1*, *ACAN*) and catabolic genes (*MMP13* and *ADAMTS5*) was evaluated. Following IL-1β treatment, anabolic gene expression was decreased relative to untreated controls (Figure 2B–D). Additionally, catabolic gene expression, e.g., *MMP13* was found to have increased following IL-1β treatment in both 2D and 3D cultures (Figure 2E,F). The expression of anabolic genes in 3D cultures treated with TGF- β was increased (Figure 2B–D), however, the expected increase in anabolic gene expression in 2D cultures was modest or not detectable (Figure 2B–D). Unexpectedly, TGF-β treatment promoted catabolic gene induction in 3D cultures, with significant increases in *MMP13* (Figure 2E) and *ADAMTS5* expression (Figure 2F).

### 2.3. Relationship between Chondrogenic Gene Expression and EV Release

We also investigated the impact of media supplements on chondrocyte EV release by comparing an EV-depleted FBS growth media to one that promotes and maintains chondrogenic differentiation in culture (media containing an insulin–transferrin–selenium (ITS) supplement). Indeed, when comparing anabolic gene expression in chondrocytes in 2D culture, ITS supplementation had greater chondrogenic gene expression than those supplemented with FBS (Figure 3A–C). Similarly, 3D cell cultures with ITS displayed increased *COL2A1* and *ACAN* expression compared to culture in FBS, whereas SOX9 expression was elevated in 3D culture conditions regardless of the type of media supplementation (Figure 3A–C). Regarding EV yields from chondrocytes under these culture conditions, the concentration of EVs was significantly greater from cells cultured in ITS compared to those in FBS when grown in 2D monolayers. Aligned with our previous experiments, greater concentrations of EVs were detected when cells were grown in 3D pellet culture and this was irrespective of the media conditions (Figure 3D). These data therefore suggest that chondrocyte EV yield is linked to the extent of chondrogenic differentiation and thus, by promoting greater differentiation, the parent cells release more EVs (Figure 3D).

Having shown that specific culture conditions increased both chondrogenic gene expression and EV yield, we next explored whether these conditions altered the subsequent phenotype of the respective EVs. To investigate this, chondrocytes cultured in 2D monolayers with EV-depleted FBS were treated with EVs isolated from chondrocytes grown in 2D or 3D (Figure 3E). Interestingly, only EVs derived from 3D cultures significantly increased the expression of *SOX9* and *ACAN* and led to a three-fold increase in *COL2A1* expression (Figure 3E–G). This may represent a homeostatic feedback response of chondrocytes to promote enhanced chondrogenesis. Interestingly, we also detected a significantly greater number of newly generated EVs measured in the supernatants of chondrocytes following treatment with EVs from 3D cultures when compared to those from 2D cultures (Appendix A).

### 2.4. Chondrocyte EV Uptake

To shed further light on the different phenotype of 2D- and 3D-derived EVs, their uptake by chondrocytes was investigated. Chondrocytes were cultured with media supplemented with ITS or FBS and incubated with fluorescently labelled EVs. After four hours, cells were washed, fixed and imaged via confocal microscopy. EV uptake was quantified by analysis of the mean grey value of BODIPY–Texas red (BODIPY-TR) relative to the number of cells per field of view. Representative images are shown in Figure 4A. Regardless of culture conditions (FBS or ITS), 3D-derived EVs were taken up by recipient chondrocytes to a significantly greater extent than 2D-derived EVs (Figure 4B). Furthermore, their uptake was significantly greater under ITS conditions versus FBS conditions (Figure 4B), suggesting that chondrocyte EV uptake is modulated by chondrogenic differentiation.

### 2.5. Role of β1-Integrin in Chondrocyte-EV Uptake

It was previously shown that the β1-integrin is the most abundantly expressed β-integrin on chondrocytes [21]. We hypothesised that the β1-integrin was also expressed on chondrocyte EVs and may play a role in EV uptake. To address this hypothesis, the expression of β1-integrin on 3D-chondrocyte-derived EVs was quantified using the nano-flow cytometer. Accordingly, the β1-integrin was detected on CFSE^+^ 3D EVs, with a frequency of expression of ~62% ± 4%. Figure 5A illustrates the gating strategy and provides representative histograms for the staining of the β1-integrin on CFSE-stained 3D EVs compared to the isotype control. To investigate the role of the β1-integrin in the uptake of EVs by chondrocytes, 3D-EVs were incubated with a β1-integrin blocking antibody prior to treatment. Representative images of BODIPY-TR-labelled 3D EVs that were untreated (Figure 5B), or treated with mouse IgG (Figure 5C) or with a β1-integrin-blocking antibody (Figure 5D), demonstrate a reduction in labelled 3D EV uptake upon β1-integrin blockade. The quantification of labelled 3D EV uptake confirms that treatment with a β1-integrin blocking antibody significantly reduced the uptake of fluorescently labelled 3D EVs by recipient cells when compared to 3D EVs incubated with an isotype control (Figure 5E). We next assessed whether EV uptake altered the chondrocyte cytoskeleton by monitoring F-actin with phalloidin staining. The addition of 3D-derived EVs to chondrocytes did not alter the total cytoskeletal levels of F-actin compared with vehicle control (Appendix A).

Collectively, the data herein demonstrate that chondrocyte-derived EVs are sequestered within the cartilage ECM, that there is a bi-directional relationship between chondrocyte EV release and the extent of chondrogenic differentiation and, finally, that the uptake of chondro-EVs is partially dependent on the β1-integrin.

## 3. Discussion

We report here the first characterization of EVs isolated from the matrix of the aggregate 3D chondrocyte cultures. Previous studies have characterized EVs from chondrocyte supernatants. As anticipated, from our data, these supernatant EVs had very little effect [22]. Herein, we isolated EVs from 3D chondrocyte cultures via mechanical disruption and, using a variety of techniques, we validated these EVs and their sequestration within the cartilage ECM. There is not a wealth of data in this area. Notably, Rosenthal et al. analysed the proteome of articular cartilage vesicles between healthy and osteoarthritic cartilage, by both mechanical disruption and enzymatic digestion of the cartilage, to obtain EVs [17].

The culture of cells in 3D-utilizing scaffolds and bioreactors is known to boost EV yield and functionality [23]; we found this to be the case when generating EVs from chondrocytes in 3D pellet culture. While it is known that EVs isolated from chondrocytes are pro-chondrogenic [8,15], we provide novel evidence that chondrocyte EVs from a 3D culture exert a far more potent pro-chondrogenic effect than those from 2D monolayers. Hence, this is the first report pertaining to the augmenting properties of 3D EVs on chondrocytes. Similar findings have been observed with skeletal muscle cells cultured in 3D, which had a greater expression of markers indicative of differentiation and maturation compared to those cultured in 2D [24], as well as osteoblasts and MSCs [25,26]. Indeed, MSCs cultured in 3D have been found to generate EVs with far greater neuronal growth properties [27]. It has also been shown that EVs generated from 3D cultures replicate the cargo profile of EVs in vivo more faithfully than those isolated from 2D cells. Furthermore, Jalilian et al. demonstrated an increased yield of EVs isolated from MSCs in 3D culture conditions [27]. Indeed, we similarly demonstrated an increase in EV yield from chondrocytes in 3D versus 2D. Other reports have utilised 3D cell growth as a means of increasing EV yield from cells towards a therapeutic goal [28].

How chondrogenic differentiation affects EV yield has rarely been studied. Rosenthal et al. demonstrated that the treatment of chondrocytes with rapamycin induces increased EV release, which was attributed to a modulatory effect on autophagy [29]. We reason that the rapamycin-induced bolstering of EV production might be an effect of increased chondrogenic gene expression. Indeed, there is evidence of rapamycin not only promoting chondrogenic gene expression but offering protective benefits with regards to cartilage degradation during OA [30]. We found that 2D chondrocytes cultured in ITS have greater EV output than those cultured with FBS. Similarly, the EV yield in cells grown in 3D is greater than that of 2D culture. Both methods of promoting greater chondrogenic gene expression also increase chondrocyte EV yield, which suggests the existence of a feedback loop between the two processes. Further evidence for this functional relationship was observed when chondrocytes were treated with cytokines. Unexpectedly, we observed an increase in catabolic gene expression in 3D cultures treated with TGF-β; however, the current literature indicates that this response is not completely unexpected, as it demonstrates a remodelling phenotype. Plaas et al. demonstrated that the synergetic increase in both TGF-β and ADAMTS5 are essential for efficient pericellular matrix turnover [31]. Given the increased proteoglycan expression in TGF-β treated cultures, as seen by alcian blue staining (Appendix A), we believe this is also the case in our experimental conditions. Whilst 24 h of IL-1β treatment of 2D and 3D cells had no effect on EV yield, an increase was observed in 3D cultures treated with TGF-β; a well-studied potent promoter of chondrogenesis. The importance of TGF-β is seen in conditional knockouts of *tgfβr2* in articular chondrocytes as well as proteins downstream of TGF-βR2, which causes an OA-like phenotype in these mice [32,33]. Notably, genome-wide association studies in humans have revealed links between the TGF-β superfamily and OA [34]. As we have shown, cues which promote chondrogenic gene expression promote an increased EV yield from chondrocytes, which includes TGF-β. Interestingly, this was only observed in 3D cells: we cannot exclude higher receptor levels or signalling clusters in these culture settings. Our results indicate that the production of EVs is matched to the extent of cartilage matrix, which is dictated by pro- and anti-chondrogenic conditions, such as TGF-β stimulation vs. the catabolic stimulus of IL-1β. In this instance 24 h TGF-β treatment did not directly increase EV release in 2D monolayers. The increased yield of EVs from mechanically dissociated 3D cultures treated with TGF-β is likely related to the greater ECM accumulation in these cultures, as observed with Alcian blue staining (Appendix A). This is in line with the current body of work, which demonstrates that chondrocyte EVs are sequestered within the cartilage matrix. Therefore, not only do we report that large amount of EVs remain sequestered within cartilage ECM, but we propose that they represent a previously underappreciated yet functional cartilage component.

We have also observed an integral role of β1-integrin in the uptake of chondrocyte-EVs. β1-integrin is the most predominantly expressed integrin in chondrocytes and it is essential in supporting the focal adhesion of these cells to cartilage components. Through this action, as part of the αβ-integrin heterodimers, β1-integrin promotes chondrocyte survival, differentiation and proliferation, inhibits apoptosis and provides cues which regulate matrix remodelling [35]. A large range of αβ1-integrin ligands exist; for example, the matrix protein fibronectin binds α3β1, α4β1, α5β1 and αVβ1 integrins, while integrins such as α1β1, α2β1, α10β1 and α11β1 display preferential binding to type II collagen [36]. Importantly, integrins mediate the adhesion of chondrocytes with the rich ECM surrounding them. Integrin activity is also involved in converting the mechanical forces applied to the cartilage into cellular signalling responses in chondrocytes. Periodical mechanical loads facilitate the fibronectin-α5β1 bond. Periodical mechanical forces promote the downstream activation of protein kinase C signalling and cause hyperpolarisation of the chondrocyte membrane. A variety of β1-mediated mechanotransducive signalling pathways have been reported to date. These include β1-Src-GIT ArfGAP1 (GIT1) and β1-Ca2+/calmodulin-dependent protein kinase II (CaMKII) signalling pathways. Functionally, mechanical inputs via integrins are responsible for chondrocyte proliferation and matrix synthesis. Alongside traditional ‘outside-in’ signalling mediated by integrins, these molecules are subject to ‘inside-out’ signalling, which can alter the binding affinity, ability to cluster and combination of integrin dimers, which allows for the dynamic regulation of chondrocyte interactions with the ECM. During OA, changes to the components and structure of cartilage lead to alterations in the integrin-mediated signalling pathways inducing the increased production of cartilage-degrading molecules, such as ADAMTS-5 and MMP-13 [37]. Given the vital role of β1-integrin in the adhesion of chondrocytes, we wondered whether chondrocyte-derived EVs shared the same dependency on β1-integrin for adhesion to matrix components and, as a result, uptake by recipient cells.

To validate this hypothesis, firstly, we confirmed the expression of β1-integrin on 3D EVs using nano-flow cytometry and, secondly, we demonstrated a reduction in 3D EV uptake after β1-integrin blockade. The involvement of β1-integrin is also functionally present when we investigated the level of EV uptake among cells cultured in FBS vs. ITS. The uptake of these EVs is greater in cells supplemented with ITS. As shown, the adhesion of these EVs is reliant on the expression of β1-integrin and its interactions with matrix components. Therefore, cells with greater chondrogenic gene expression, and hence, greater cartilage gene expression, display greater EV adhesion and uptake. However, despite the importance of the β1-integrin in focal adhesion, the binding of 3D-derived EVs to chondrocytes did not alter the cytoskeletal levels of F-actin, as quantified with phalloidin staining (Appendix A). Interestingly, there was an increase in the total number of nuclei per field of view following 24 h of EV treatment. These data suggest an increase in the proliferation of chondrocytes treated with 3D-derived EVs. These data further support the involvement of β1-integrin in the uptake of these EVs, as it has previously been shown that the activation of the β1-integrin on chondrocytes promotes proliferation [35]. However, further work is required to elucidate the exact mechanism by which this occurs. These data substantiate the notion that chondrocyte EVs are sequestered within the cartilage via interactions with adhesion molecules and matrix components. These interactions are key in mediating the uptake of these chondrogenic EVs and vital for maintaining chondrocyte and overall cartilage health.

Through a comparison of the conditions in which we cultured chondrocytes, we can create a picture of how the chondrocyte–EV–cartilage axis may become perturbed in a chronic degenerative disease like OA. It can be speculated that, in diseased cartilage, the number of EVs is reduced following chondrocyte death. Of the remaining chondrocytes, alterations in the phenotype away from homeostasis further reduce EV production and the chondrogenic potency of those EVs is also reduced, much like the changes which occur between cells cultured in FBS or ITS. It is also likely that changes to the make-up and biomechanical properties of the diseased cartilage further hinder the ability to generate and uptake EVs. Over a substantial period of time, these changes contribute to a negative feedback loop, further exacerbating pathogenicity within the cartilage.

In the context of OA, cartilage degradation and chondrocyte death lead to the production of molecules which reduce chondrogenic gene expression across the chondrocyte population. These further feed back into a decrease in chondrogenic gene expression [3]. These data suggest that paracrine communication among chondrocytes via EVs is key in this process. Chondrocytes with reduced chondrogenic gene expression produce a lower number of EVs and the chondrogenic potential of these EVs is also reduced. Others have used a combination of biologically existing EVs, either in combination with additional synthetic payloads or engineered to deliver a variety of proteins and nucleic acids [8]. Others have used autologous-plasma-derived EVs to promote chondrogenesis in vitro. Some efficacy was demonstrated for autologous plasma EVs in a small clinical trial for temporal bone cavity inflammation, as well as a variety of pre-clinical models. Chondrocyte EV treatment within the joint presents a novel modality of promoting cartilage healing in patients. Indeed, pre-clinical models utilising chondrocyte EVs as a treatment strategy have shown efficacy in alleviating cartilage degradation [38]. Another interesting application of the work herein is in the use of hydrogels. Hydrogels can be used as carriers for drug transportation and can be implanted into the affected area by injection and surgery, improving the efficacy of the given drug. Hydrogels can also be used as scaffolds to mediate the adhesion, proliferation and differentiation of stem cells in cell-based therapies [39]. Curiously, in either of these conditions, this current body of work suggests that combining such hydrogels with chondrocyte EVs would bolster their chondroprotective effects in OA. However, a more detailed understanding of the biology governing chondrocyte EV migration and uptake and how they influence cartilage health is required before novel EV-based therapies can be designed for arthropathies.

## 4. Materials and Methods

### 4.1. C28/I2 Chondrocyte Cell Line Culture

C28/I2 human chondrocyte cell lines were obtained with thanks from Prof. Mary Goldring’s Lab. C28/I2 was grown in complete media comprising Dulbecco’s Modified Eagle Medium/1:1 Hams F-12 Nutrient Mixture (DMEM/F-12) with GlutMAX (Thermofisher Scientific, Cambridge, UK) +10% non-heat-inactivated FBS (Thermofisher Scientific Cambridge, UK) with 1% penicillin and streptomyosin (P/S; Sigma-Aldrich, Gillingham, UK).

### 4.2. 2D and 3D Formats of Chondrocyte Culture

C28/I2 cells at 80% confluence were detached from T75 flasks following trypsinisation, as described above. Chondrocyte monolayers were cultured by adding 3 × 10^5^ chondrocytes to a 6-well plate in complete media for 24 h. Pellet cultures were generated by taking 3 × 10^5^ chondrocytes in 1 mL of complete media and adding the cell suspension to a 15 mL falcon before centrifuging at 300× *g* for 5 min to pellet the cells, and then placing this upright for 24 h with lids untightened to allow for adequate gaseous exchange.

For all cultures, complete media was removed after 24 h and the cultures were washed with DPBS^−/−^. Following this, serum-free and phenol-red-free DMEM/F-12 (Thermofisher Scientific, Cambridge, UK) with 1:100 Insulin-Transferrin-Selenium (ITS; Sigma-Aldrich, Gillingham, UK) and 1% P/S was added and cultures were incubated for a further 24 h. All cultures were stimulated as detailed below. In some experiments, the last 48 h of culturing was carried out in conditions containing serum; in these experiments, phenol-red-free DMEM/F-12 with 10% FBS, exosome-depleted (Thermofisher Scientific), and 1% P/S was used instead. All incubations of cultured cells were carried out at 37 °C with 5% CO_2_.

### 4.3. C28/I2 Chondrocyte Stimulation

Chondrocytes were treated in serum-free chondrocyte media for 24 h at 37 °C with either IL-1β (30 ng/mL; Peprotech, Cheshire, UK), TGF-β (10 ng/mL; Peprotech, Cheshire, UK) or vehicle only.

### 4.4. Extracellular Vesicle Isolation and Staining

Cell culture supernatants used for extracellular vesicle isolation were centrifuged at 4400× *g* for 15 min to pellet and remove cells. The resulting supernatant underwent two rounds of centrifugation for 2.5 min at 13,000× *g* to pellet and remove cellular debris. Finally, EVs were pelleted at 20,000× *g* for 30 min and the supernatant discarded. The resultant EVs were resuspended in 0.22 μm double-filtered DPBS^−/−^ (DF DPBS^−/−^) for staining. EVs were stained with either BODIPY-FL (1.67 µM; Invitrogen, Paisley, UK), CFSE (1.67 nM; Invitrogen, Paisley, UK) or BODIPY-Texas red (50 µM; Invitrogen, Paisley, UK) at 4 °C for 30 min. Following staining, EVs were washed in DF PBS^−/−^ for 30 min at 20,000× *g*. Pelleted EVs were re-suspended in DF DPBS^−/−^ and quantified as described below. All centrifugation steps were carried out at 4 °C.

For β1-integrin expression, EVs were incubated with 4 µg/mL mouse anti-human β1-integrin APC (Clone:TS2/16; eBioscience, Altricham, UK) or IgG1 isotype control (Clone: RMG1-1; Invitrogen, Paisley, UK) in conjunction with either CFSE or BODIPY-Texas Red at 4 °C for 30 min. EVs were then washed and resuspended as described above and analysed on the nano-flow cytometer (NanoFCM, Nottingham, UK). For experiments requiring the functional blocking of β1-integrin, 15 µg/mL of mouse monoclonal anti-human β1-integrin (Clone: P5D2; R&D Systems, Minneapolis, MN, USA) or 15 µg/mL of mouse IgG1 isotype control (Clone:11711; R&D Systems, Minneapolis, MN, USA) were used.

### 4.5. NanoFCM Quantification and Analysis

Prior to use, the NanoFCM was calibrated using dual-laser quality control beads (NanoFCM) to standardise laser settings between each use of the instrument. Quality control beads at a fixed nanoparticle concentration per ml were used to establish the nanoparticle concentration of a sample with unknown concentration. EV size was established based on a standard curve generated by running Silica Nanospheres Cocktail #1 (NanoFCM) at a range of known sizes, 65–155 nm. After laser, size and concentration standardisation, an aliquot of DF DPBS^−/−^ lacking EVs was acquired as a blank to remove background nanoparticles from EV samples. Fluorescently labelled EVs were acquired on the nanoFCM with a sampling pressure of 1.0 kPa and events were recorded for a total of 1 min.

To ascertain a more robust evaluation of EV concentration, following acquisition, CFSE-stained samples were treated with 0.01% triton x-100 (Sigma-Aldrich) at room temperature for 20 min and run again on the nanoFCM. Data showing nanoparticle concentrations for EVs stained with CFSE were calculated by subtracting CFSE and triton EV concentrations from CFSE-only EV concentrations.

### 4.6. Transmission Electron Microscopy

Electron microscopy (EM) was performed at the Royal Brompton hospital on EV samples from chondrocytes either in monoculture or pellet cultures with or without triton treatment (as previously described). Briefly, EM 400 mesh copper grids were pre-coated with 1% formvar solution prepared in chloroform. The EV suspension was directly added onto the top side of EM grids and EVs were left to adhere for 5–10 min. EM grids were washed twice with filtered distilled water, fixed with 1% glutaraldehyde in PBS and finally stained in 2% uranyl acetate for 2 min. Excess stain was absorbed using filter paper and grids were left to dry in a closed container. A JEOL 1400+ TEM (Welwyn Garden City, UK) was used to collect a series of images from the EM grids at different magnifications, and 4 K × 4 K digital images at between 8000x and 20,000x were captured using AMT digital camera software (AMT 16X, Deben UK Ltd.; Bury St. Edmunds, UK).

### 4.7. Western Blotting

EVs were isolated from 3D cultures as described above. EVs and cells were lysed by adding 20 μL or 100 μL RIPA buffer (Thermofisher Scientific, Cambridge, UK) with 1% protease inhibitor (Thermofisher Scientific, Cambridge, UK) respectively. Total protein was quantified using a microBCA assay (Boster Bio, Pleasanton, USA) according to the manufacturer’s instructions. A total of 13 μg (EVs) or 30 μg (cells) of total protein was added to laemmeli buffer (Thermofisher Scientific, Cambridge, UK) with 1% *v*/*v* Dithiolthereitol (0.1 M DTT) and heated at 95 °C for 10 min. Samples were loaded onto a 10% polyacrylamide NuPAGE™ mini protein gel (Thermofisher Scientific) alongside the Spectra Multicolor Broad Range protein ladder to determine molecular weight (Thermofisher Scientific). Gel electrophoresis was carried out at 60 V for 20 min and then at 120 V for 60 min in 1x NuPAGE™ MOPS SDS Running Buffer (Thermofisher Scientific, Cambridge, UK). Gels were transferred onto an Immobilon^®^-E PVDF Membrane (Merck, Darmstadt, Germany) using a wet transfer technique at 90 V for 90 min in 1x NuPAGE™ Transfer Buffer (Thermofisher Scientific, Cambridge, UK).

The membrane was blocked in 5% *w*/*v* BSA PBS-T while shaking for 1 h at room temperature and then briefly washed with 0.1% BSA PBS-T. Primary antibodies (see Table 1) were made in 1% *w*/*v* BSA PBS-T and membrane was incubated overnight on a shaker at 4 °C. Following incubation, the membrane was washed three times in PBS-T and appropriate secondary horseradish peroxidase (HRP)-conjugated antibody (see Table 1) was added at room temperature for 1 h on a rocking shaker. The membrane was washed three times with 1% *w*/*v* milk in PBS-T for 10 min each, after which Pierce™ ECL Western Blotting Substrate (Thermofisher Scientific, Cambridge, UK) was added to develop the blot. Images were captured in a Gel Doc system (FluorChem E). After probing for calnexin, membranes were incubated in Restore™ Western Blot Stripping Buffer (Thermofisher Scientific, Cambridge, UK) for 15 min at room temperature on a rocking shaker and then washed 3 times in PBS-T for 10 min each. Following washing, the process of blocking, staining and developing the membrane was repeated for CD63 expression.

### 4.8. Extracellular Vesicle Uptake Assay

Unless otherwise specified, chondrocyte EV uptake assays were carried out as follows. C28/I2 chondrocytes were cultured in monolayers to 80% confluency in IBIDI µ-slide 8-well high chamber slides (Ibidi^®^, Gräfelfing, Germany). EVs labelled with BODIPY-Texas-red were added to the monolayers for four hours at a ratio of 6:1 EVs per chondrocyte, all incubations of cultured cells were carried out at 37 °C with 5% CO_2_ in the dark. Cells were washed after four hours with DF PBS and fixed with 4% PFA for 15 min. Fluoroshield™ with DAPI (Sigma-Aldrich, Gillingham, UK) was added to each condition prior to imaging via confocal microscopy on the Ziess LSM 800 (Ziess, Oberkochen, Germany). Uptake was quantified by mean grey value of BODIPY-Texas red per cell as an average of four fields of view per condition. Experiments requiring the functional blockade of β1-integrin were performed by treating EVs with anti-β1-integrin antibody and washing prior to incubation with chondrocyte monolayers. For experiments requiring cytoskeletal staining, following fixation with 4% PFA and permeabilization with 1% TritonX-100, cells either treated with BODIPY-TR EVs or not were incubated with FITC-conjugated Phalloidin (ThermoFisher Scientific, Cambridge, UK) as per manufacturers instructions. After the cells were washed, Fluoroshield™ with DAPI (Sigma-Aldrich, Gillingham, UK) was added and imaged as stated above.

### 4.9. mRNA Analysis

Chondrocyte RNA was isolated using the RNeasy mini kit (Qiagen, Manchester, UK) as per the manufacturer’s instructions. RNA yield was quantified on the Nanodrop ND-1000, and 1 ng of total RNA was used to generate cDNA using the RevertAID RT reverse transcription kit (ThermoFisher Scientific, Cambridge, UK) as per the manufacturer’s recommendations. For each PCR reaction, a total volume of 10 μL was used, consisting of 1 μL of cDNA, 5 μL of 2X Power SYBR Green Mastermix (Applied Biosystems, Inc., Norwalk, USA), 3 μL of dH_2_O, and 1 μL of gene-specific primer. Primers were commercially available; the probes for specific mRNA targets can be found in Table 2. Realtime PCR was carried out with the following amplification settings: 95 °C for 15 min; 40 cycles of 95 °C for 15 s, 60 °C for 1 min using the CFX connect real-time PCR system (Bio-Rad Laboratories Ltd., Watford, UK). Data are expressed as relative units calculated by 2^−Δ∆CT^ by normalization relative to 18 s and to fold change over untreated controls.

### 4.10. Alcian Blue Staining

3D-cultured chondrocytes were generated as described above, at the end of which 3D cultures were fixed in ice cold methanol (Sigma-Aldrich, Gillingham, UK) at −20 °C overnight. 3D cultures were then washed in warm PBS^−/−^ and stained in Alcian blue 8GX solution (1% in 0.1 N hydrochloric acid; pH 1.0; Atom Scientific, Hyde, UK) overnight. Cultures were then rinsed in 0.1 N hydrochloric acid (Sigma-Aldrich, Gillingham, UK), an 8 M Guanidine–HCL solution (ThermoFisher Scientific, Cambridge, UK) was added and the samples were left on a mild shaker overnight. The optical density of the solution was measured at 630 nm on a microplate reader. DNA content was quantified by staining double-stranded DNA using Quant-it PicoGreen dye (ThermoFisher Scientific, Cambridge, UK), as directed by the manufacturers. Fluorescence was measured at an excitation of 480 nm on a NOVOstar microplate fluorometer (BMG Lab technologies, Aylesbury, UK). Alcian blue optical density was normalised to double-stranded DNA fluorescence and the values plotted as percentage ECM accumulation relative to control.

### 4.11. Statistical Analysis

All statistical analyses were conducted on GraphPad Prism 9. All statistical parameters and relevant statistical tests are described in figure legends. Data depict mean ± SEM, with *p* < 0.05 considered to be statistically significant. For all experiments, the value of n is represented by points within bars and is documented within the figure legend; this value represents the number of biological replicates.

## Figures and Tables

**Figure 1 ijms-25-04756-f001:**
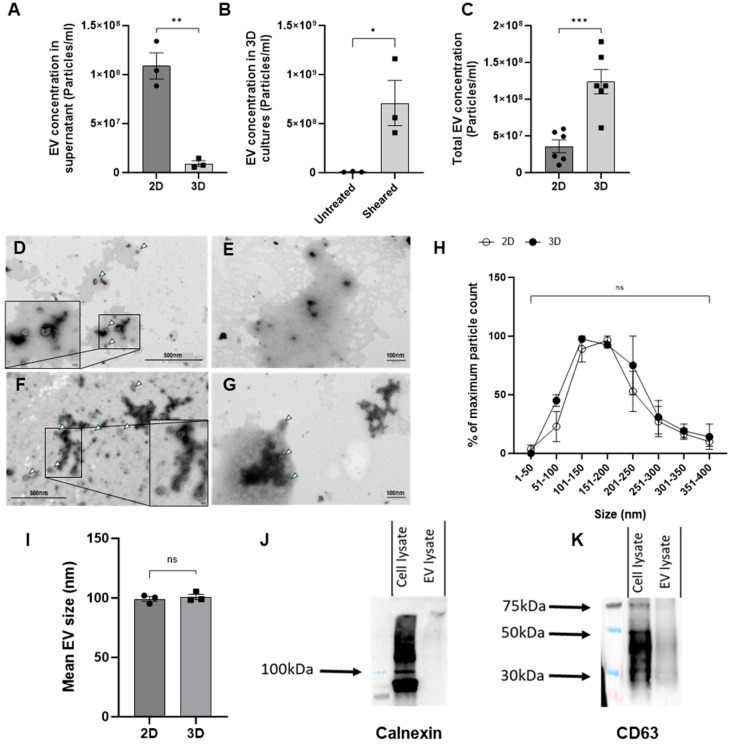
**Characterisation of chondrocyte-derived EVs.** (**A**) EVs isolated from the supernatants of 2D- or 3D-cultured chondrocytes were quantified by nano-flow cytometry. (**B**) EV concentrations determined from EVs isolated from 3D cultures, either untreated or following the mechanical dissociation of 3D cultures (Sheared). (**C**) Comparison of EV concentrations isolated from either supernatants of 2D monolayers or 3D cultures following mechanical dissociation. (**D**–**G**) Representative TEM images of EVs (white arrowheads) isolated from (**D**) 2D supernatant (**E**) 2D supernatant with 0.01% triton-X F. 3D homogenate and (**G**) 3D homogenate with 0.01% triton-X. (**H**) Quantification of EV size distribution isolated from 2D supernatant or 3D sheared pellet normalised to the maximum particle count per sample. (**I**) Mean size of EVs from 2D or 3D sources as measured by nano-flow cytometry. (**J**) Western blot analysis of Calnexin expression in either 3D C28/I2 cell lysates or EVs. (**K**) Western blot analysis of CD63 expression in either 3D C28/I2 cell lysates or EVs. n = 1 of 5 pooled samples. Data represent the mean ± SEM (**A**) n = 3; unpaired *t*-test. (**B**) n = 3; unpaired T-test. (**C**) n = 4–6; unpaired *t*-test. (**H**) Each point represents n = 2 biological replicates made up of 15 fields of view each; two-way ANOVA followed by Šídák’s multiple comparisons test. (**I**) n = 3; unpaired *t*-test (ns = *p* > 0.05, * = *p* < 0.05, ** = *p* < 0.01 *** = *p* < 0.001). Abbreviations: EV: extracellular vesicle.

**Figure 2 ijms-25-04756-f002:**
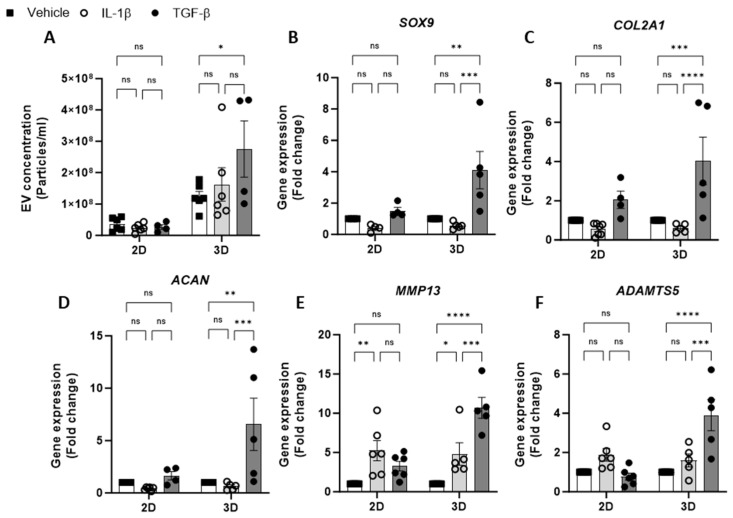
**Effect of cytokine stimulation on chondrocyte EV release.** (**A**). Concentration of EVs isolated from 2D and 3D chondrocytes treated with either vehicle, IL-1β (30 ng/mL) or TGF-β (10 ng/mL) for 24 h. Relative gene expression of (**B**) *SOX9*, (**C**) *COL2A1*, (**D**) *ACAN*, (**E**) *MMP13* and (**F**) *ADAMTS5* in 2D and 3D chondrocyte cultures treated with either vehicle, IL-1β (30 ng/mL) or TGF-β (10 ng/mL) for 24 h. Data represent the mean ± SEM (n = 4–6); (**A**–**F**) Two-way ANOVA with Turkey’s multiple comparison test; (ns = *p* > 0.05, * = *p* < 0.05, ** = *p* < 0.01, *** = *p* < 0.001, ****= *p* < 0.0001). Abbreviations: IL-1β: interleukin-1 beta, TGF**-**β: transforming growth factor beta, EV: extracellular vesicle, SOX9: SRY-box transcription factor 9, COL2A1: collagen type II alpha1, ACAN: aggrecan, MMP13: matrix metalloproteinase-13, ADAMTS5: A disintegrin and metalloproteinase with thrombospondin motifs 5.

**Figure 3 ijms-25-04756-f003:**
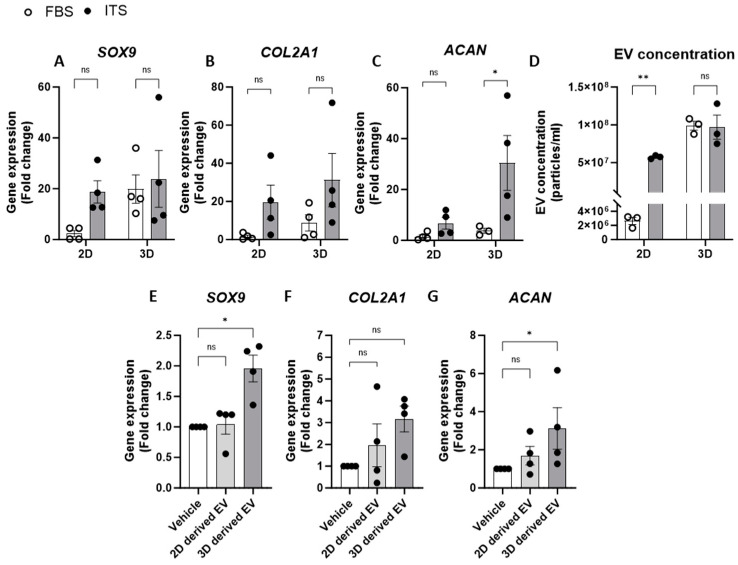
**Relationship between chondrogenic gene expression and EV release.** (**A**–**C**) Gene expression analysis of chondrogenic genes, (**A**) SOX9, (**B**) COL2A1 and (**C**) ACAN, from 2D or 3D chondrocytes supplemented with either FBS or ITS. (**D**) Concentration of EVs from chondrocytes in 2D or 3D generated in media supplemented with FBS or ITS. Gene expression of (**E**) SOX9, (**F**) COL2A1 and (**G**) ACAN in 2D following 24 h treatment with EVs isolated from 2D or 3D chondrocytes. Data represent the mean ± SEM (n = 4); (**A**–**D**) Two-way ANOVA with Sidaks multiple comparison, (**E**–**G**) Kruskal–Wallis with Dunn’s multiple comparison. (ns = *p* > 0.05, * = *p* < 0.05, ** = *p* < 0.01). Abbreviations: FBS: fetal bovine serum, ITS: insulin-transferrin-selenium, SOX9: SRY-box transcription factor 9, COL2A1: collagen type II alpha1, ACAN: aggrecan, EV: extracellular vesicle.

**Figure 4 ijms-25-04756-f004:**
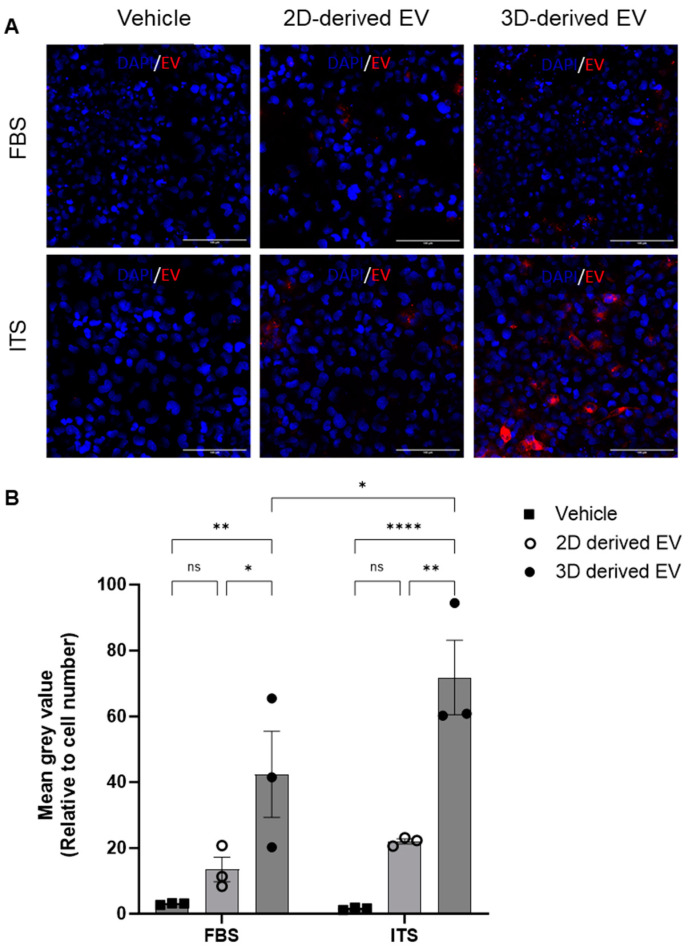
**Uptake of chondrocyte-derived EVs.** EVs were collected from either 2D or 3D chondrocyte cultures, labelled with BODIPY Texas-red and chondrocyte uptake determined by immunofluorescence after 4 h. (**A**) Representative images of Texas-red BODIPY-positive EV uptake. Chondrocytes were grown in monolayers with media supplemented with either FBS or ITS and the subsequent uptake of EVs was assessed (red) with DAPI nuclear staining (blue). (**B**) Quantitative assessment of EV uptake; Texas-red BODIPY mean grey value normalised to cell number. Data represent the mean ± SEM (n = 3); (**B**) two-way ANOVA with Sidak’s multiple comparison. (ns = *p* > 0.05, * = *p* < 0.05, ** = *p* < 0.01, **** = *p* < 0.0001). Abbreviations: EV: extracellular vesicle, FBS: fetal bovine serum, ITS: insulin-transferrin-selenium.

**Figure 5 ijms-25-04756-f005:**
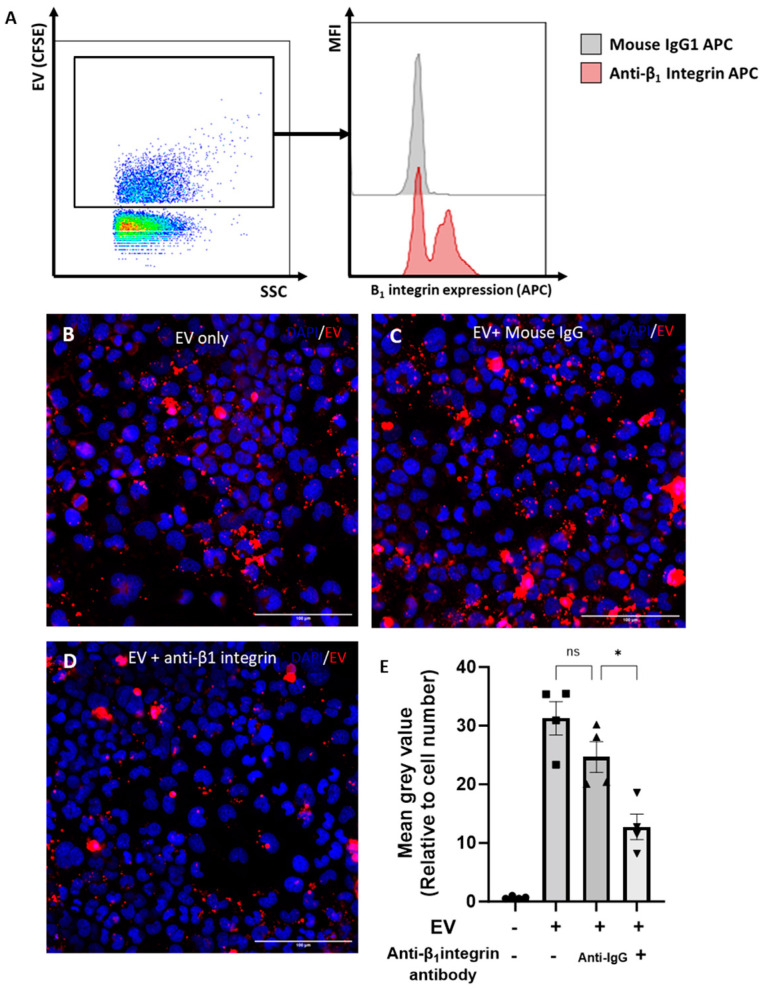
**Role of β1-integrin in chondrocyte EV uptake.** (**A**) Expression of β1-integrin on chondrocyte EVs by nano-flow cytometry. EVs were gated by selection for positive staining for CFSE (FITC). Representative histogram of β1-integrin staining (red) compared to isotype control (grey). (**B**–**D**) Uptake of chondrocyte-derived EVs, with or without pre-incubation with an isotype control or blocking β1-integrin antibody (15μg/mL), as assessed by immunofluoresence. Chondrocytes were grown in monolayers (with media supplemented with ITS) and incubated for 4 h with BODIPY-TR labelled EVs (red; from 3D cultures). (**E**) Quantitative assessment of EV uptake; BODIPY-TR mean grey value normalised to cell number. Data represent the mean ± SEM (n = 4) (**E**) One-way ANOVA with Turkey’s multiple comparison test (ns = *p* > 0.05, * = *p* < 0.05).

**Table 1 ijms-25-04756-t001:** Antibody list for Western blotting.

Antibody (Clone)	Manufacturer	Catalogue Number	Dilution Factor
Calnexin Monoclonal Antibody (GT1563)	ThermoFisher Scientific	MA5-31501	1:1000
CD63 Recombinant Rabbit Monoclonal Antibody (2H5I1)	ThermoFisher Scientific	MA5-35208	1:1000
Goat anti-mouse IgG1-HRP	Southern Biotech	1071-05	1:2000
Polyclonal Goat Anti-Rabbit immunoglobulin-HRP	Dako	P0448	1:2000

**Table 2 ijms-25-04756-t002:** Polymerase chain reaction-specific mRNA probes.

PCR Probe	GeneGlobe ID	Manufacturer
*COL2A1*	QT00019544	Qiagen
*ACAN*	QT00996618	
*SOX9*	QT00001498	
*MMP13*	QT00001764	
*ADAMTS5*	QT00011088	
*18S*	QT00199367	

## Data Availability

The original contributions presented in the study are included in the article/Appendix A. The raw data supporting the conclusions of this article will be made available by the authors on request. Further inquiries can be directed to the corresponding author.

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
