# Peer review of "β1-Integrin-Mediated Uptake of Chondrocyte Extracellular Vesicles Regulates Chondrocyte Homeostasis"

_ijms, 2024, doi:10.3390/ijms25094756_

Round 1

Reviewer 1 Report

Comments and Suggestions for Authors

The manuscript presents interesting data on the role of chondrocyte-derived EVs in a 3D culture model. However, several key issues need to be addressed before the manuscript can be considered for publication:

TGF-β treatment increased the yield of EVs from 3D cultures but not from 2D cultures, and unexpectedly also induced the expression of catabolic genes (MMP13 and ADAMTS5) in these cultures. Can the authors comment on the mechanism(s) by which TGF-β treatment increases EV release only in 3D cultures? Is this related to the increased presence of ECM or other factors specific to the 3D microenvironment?

The authors observed that TGF-β treatment increased the expression of catabolic genes alongside the increase in EV release in 3D cultures. Are these EVs promoting or inhibiting catabolic activity in recipient cells?

The authors claim that most EVs are sequestered within the 3D cartilage ECM. However, they don't provide convincing evidence to distinguish between EVs released from the cells and EVs trapped within the matrix from the beginning of the experiment. Did the authors perform any experiments to distinguish between these two possibilities?

The authors should discuss their findings in the context of existing literature on chondrocyte EVs and osteoarthritis (OA). How do these findings align with previous studies, and what are the potential clinical implications?

The study does not delve into the mechanisms that could be involved in the increased production of EVs in 3D cultures and with TGF-β. Has the role of the endosomal machinery in this process been evaluated?

The manuscript would benefit from a more concise and focused presentation of the results.

Reviewer 2 Report

Comments and Suggestions for Authors

The authors address a very interesting issue – the impact of 3D cell culture and paracrine signaling of chondrocytes on their function. Although the authors indicate in the title that β1 is crucial for this process their results this seem not to be strong enough to fully support such a strong anchorage of integrin in the title of the manuscript. After several updates and additional (rather simple) experiments, this work could be very interesting and beneficial in the context of understanding skeletal system homeostasis. I appreciate addressing integrin's role in this process but it should be more precisely described and investigated.

1.     I would suggest authors rewrite the abstract because it's written in a little chaotic way. For instance I would suggest changing “These EVs were assessed for their effects on chondrogenic gene expression…” to “We assessed the effect of EVs on chondrogenic…” It could make readers more interested in this work.

2.     The figure caption regarding the impact of trapping EVs in a 3d cell culture matrix seems to be a little bit confusing. I understand it after reading the text but the authors should think about more clear description also of Figure captions (especially 1A vs 1C)

3.     Could authors provide deeper discussion and more critical interpretation of TEM images of EVs?

4.     The caption of Figure 2 is bolded. Please revise if it shouldn’t be only the title of the figure caption bolded.

5.     Figure 3 D – gene name on the graph is missing

6.     Why supplementary figure 1 is located in the main text of the manuscript?

7.     Figure 4 – please precise what is marked in red.

8.     Authors assessed only one integrin – β1, while integrins are dimer proteins. It would be valuable to test particular dimers – at the list by using RGD peptide, which is biding to multiple integrins dimers (10.3390/cancers13071711) – i.e. α5β1.

9.     It should be underlined that addressing the significance of integrins appears to be an important part of this work in any case. It would be beneficial to discuss more precisely both integrins roles in chondrocytes' homeostasis as well as in general discuss the impact of the 3D condition on the function of cells responsible for skeletal system functioning (several reviews and interesting works might be found in this field i.e. 10.18388/abp.2019_2893)

10.  Authors address the role of integrins (which as underlined by authors on page 13, line 348 is crucial in focal adhesion establishment) it would be beneficial to analyze how evs affects the cytoskeleton (at least actin) organization of chondrocytes.  Such an experiment should not be too much challenging since requiring only the staining of cells with phalloidin. Several interesting quantitative parameters depicting the impact of EVs on cellular cytoskeleton organization and morphology might be obtained from such studies. 

Round 2

Reviewer 1 Report

Comments and Suggestions for Authors The authors heeded the recommendations and responded to the approaches made by this reviewer, the observations have been resolved to satisfaction.

Author Response

Many thanks for reviewing our article. 

Reviewer 2 Report

Comments and Suggestions for Authors

The authors properly addressed the majority of the Reviewers' remarks. Espetially I appreciate additional alcian blue staining, actin staining and authors consideration of future studies about other integrin subunits. The latter indicate authors awareness of complexity of addressed process. However I would suggest improving actin cytoskeleton analysis - since authors only considered fluorescence intensity - which is not an ideal marker:

We thank the reviewer for their interesting suggestion, phalloidin staining was carried out following treatment with EVs or not, data can be found in supplementary figure 3. We did not observe any changes in the amount of staining for F-actin in this experiment on a per cell basis (supplementary figure 3). The results and discussion are updated to reflect this (page 9, lines 240-242 and page 12 lines 374-377).

I would like to suggest authors to analyze also cell spreading of the cells basing on actin cytoskeleton and the number of nuclei at least.

Author Response

Thank you for the review of our article. 

Supplementary figure 3 has been updated to include analysis of the mean cell diameter and nuclei count per field of view. The discussion has been amended to reflect the inclusion of this data (Page 12 final paragraph).